# Predictive Value of Red Blood Cell Distribution Width, Mean Platelet Volume and Platelet Distribution Width in Children with Acute Appendicitis

**DOI:** 10.3390/children8111041

**Published:** 2021-11-11

**Authors:** Jelena Antić, Radoica Jokić, Svetlana Bukarica, Ivana Lukić, Dejan Dobrijević, Goran Rakić, Miloš Pajić, Veličko Trajković, Marina Milenković

**Affiliations:** 1Institute for Child and Youth Health Care of Vojvodina, 21000 Novi Sad, Serbia; jokic.rs@gmail.com (R.J.); bukaricasvetlana@gmail.com (S.B.); ivana.lukic.md@gmail.com (I.L.); dobrijevich@gmail.com (D.D.); goran.rakic@mf.uns.ac.rs (G.R.); milospajic1@gmail.com (M.P.); velickotrajkovic@gmail.com (V.T.); mdjermanov@gmail.com (M.M.); 2Faculty of Medicine, University of Novi Sad, 21000 Novi Sad, Serbia

**Keywords:** child, appendicitis, mean platelet volume, red cell distribution width, platelet distribution width, discriminant analysis, principal component analysis, analysis of variance

## Abstract

Background and Objectives: Acute appendicitis in pediatric patients is one of the most common surgical emergencies, but the early diagnosis still remains challenging. The aim of this study was to determine the predictive value of Red blood cell distribution width (RDW), Mean platelet volume (MPV) and Platelet distribution width (PDW) in children with acute appendicitis. Materials and Methods: This study was a retrospective assessment of laboratory findings (RDW, MPV, PDW) of patients who underwent surgical treatment for acute appendicitis from January 2019 to December 2020. Result: During this period, 223 appendectomies were performed at our Institute. In 107 (43%) cases appendicitis was uncomplicated, while in 116 (46.6%) it was complicated. WBC and RDW/MPV ratio were significant parameters for the diagnosis of acute appendicitis with cut-off values of 12.86 (susceptibility: 66.3%; specificity: 73.2%) and 1.64 (susceptibility: 59.8%; specificity: 71.9%), respectively. WBC and RDW/RBC ratio were independent variables for the diagnosis of complicated appendicitis. The cut-off values were 15.05 for WBC (sensitivity: 60.5%; specificity: 70.7%) and 2.5 for RDW/RBC ratio (sensitivity: 72%; specificity: 52.8%). Conclusions: WBC is an important predictor of appendicitis and complicated appendicitis. RDW, MPV and PDW alone have no diagnostic value in pediatric acute appendicitis or predicting the degree of appendix inflammation. However, the RDW/MPV ratio can be an important predictor of appendix inflammation, with higher values in patients with more severe appendix inflammation. RDW/RBC ratio may be an important predictor of complicated appendicitis.

## 1. Introduction

Acute appendicitis is a common abdominal emergency in the pediatric population all over the world. The data in the literature show that acute appendicitis is the reason for 1–2% of pediatric surgical admissions [1]. Even though the diagnosis can be challenging, most of the cases present with typical clinical findings. The rate of clinical diagnosis along with advanced imaging methods nowadays is higher than 85%, but still not completely adequate [2]. Operative management has been the gold standard of treatment for many years. Negative appendectomy still occurs in 1 to 40%, while delays in diagnosis most certainly lead to complications such as perforation, abscess formation, peritoneal inflammation, sepsis and intestinal obstruction [1,2,3]. Nonoperative treatment of uncomplicated acute appendicitis in children is an alternative to appendectomy [4]. A number of studies on adult patients which introduce new markers in the diagnosis of acute appendicitis have been conducted recently [5,6,7]. Some studies have also been conducted in the pediatric population [8,9,10,11]. Numerous parameters have been examined to support the diagnosis of acute appendicitis until today. CRP, IL-6, tumor necrosis factor, procalcitonin (PCT), granulocyte colony-stimulating factor, calprotectin, chemokine ligand-8, serum amyloid A, matrix metalloproteinase 9, and myeloperoxidase are among the inflammatory markers investigated to support diagnosis [8,12,13,14]. However, in some cases, none of the new inflammatory markers, including leukocyte, C-reactive protein (CRP), or PCT, can identify acute appendicitis with high specificity and sensitivity [13]. PCT, Interleukin (IL)-6, IL-2, and D-dimer plasma levels are elevated at different levels in the course of acute appendicitis and may be of particular value in the diagnosis of acute appendicitis [13]. Plasma fibrinogen (FB) is an acute inflammatory mediator, and its plasma level is usually raised in an acute inflammatory condition of any kind. It has been found that plasma FB is more accurate in diagnosing appendicitis and impending perforation, compared to other inflammatory markers [9].

Enterochromaffin cells that contain large amounts of serotonin are mostly located in the distal appendix. Serotonin metabolite 5-hydroxyindoleacetic acid (5-HIAA) could therefore be a marker for acute appendicitis, but it has been shown that it is not a reliable method for diagnosing acute appendicitis in children [10].

Some markers such as CRP, immature granulocyte percentage or sodium (hyponatremia) could predict complicated appendix in the pediatric population [11,15]. An elevated level of total serum bilirubin may be useful as an indicator of perforated appendicitis in children [16].

Red blood cell distribution width (RDW) is a standard component of a complete blood count (CBC) which is routinely used in emergency departments and is measured as a percentage of the number of circulating red blood cells which deviate from the mean volume. Some studies showed that elevated levels of RDW can be seen in cases of various pathological conditions such as inflammatory bowel disease, celiac disease, pulmonary embolism, coronary artery disease, as well as in patients with acute appendicitis [6,7,8]. Higher RDW may be valuable for aiding the diagnosis of acute appendicitis in children, but it is not a useful marker for predicting perforated appendicitis.

Mean platelet volume (MPV) and platelet distribution width (PDW) are also presented in the CBC. They are the indicators of platelet activation. The size of the platelet is correlated with the activity and the function of the platelet; larger platelets are more active than small ones. The values of MPV show activity of systemic inflammation. A high value of MPV can be found in chronic diseases while lower MPV values can be found in acute diseases, such as acute appendicitis. PDW is an indicator of variation in platelet size, which can be a sign of active platelet release [5,6,17,18].

PDW, RDW, and platelet values in patients diagnosed with uncomplicated appendicitis may be a guide in choosing patients to be treated with antibiotics only [17]. 

This study aimed to determine the predictive value of RDW, MPV and PDW in children with acute appendicitis and with complicated appendicitis in particular.

We hypothesized that, as an inflammatory disease of the GI tract, RDW, MPV and PDW can be used in the diagnosis of acute appendicitis in pediatric patients. The primary goal was to identify if RDW, MPV and PDW may help in the diagnosis of acute appendicitis. The secondary goal was to determine whether these parameters can help to distinguish complicated and uncomplicated forms of the disease in pediatric patients. 

## 2. Materials and Methods

This study was conducted via retrospective assessment of laboratory findings (RBC, RDW, MPV, PDW ant their ratios) of all the patients who were operated on for acute appendicitis at the Institute for Children and Youth Healthcare of Vojvodina in Novi Sad, Serbia, between January 2019 and December 2020, and had a pathology report that confirmed the diagnosis of acute appendicitis. The patients in the control group were selected from the group of those treated for nonspecific abdominal pain at our Institute.

Exclusion criteria were: patients who were treated conservatively for acute appendicitis and patients who were operated on for negative appendicitis. The study was approved by the Institutional Review Board of the Institute for Child and Youth Healthcare of Vojvodina (Number 3774-3 from 29 September 2021). The Health Information System and medical source documentation were used to retrieve the patients’ data.

The study included 223 patients below the age of 18 admitted and operated on for acute appendicitis. Blood tests of 239 patients treated at our Institute for nonspecific abdominal pain, used as a control group, were analyzed. Preoperative laboratory findings were obtained: standard complete blood count parameters including Red Blood Cell (RBC) count, Red Blood Cell Distribution Width (RDW), Mean Platelet Volume (MPV) and Platelet Distribution Width (PDW). These laboratory findings and their ratios were compared according to operative findings. The normal range of RDW is 12.0–14.0%, the normal range of MPV is 6.0–13.0 fL, and in the case of PDW it is 16.0–25.0%. The hematological values were tested on hematology analyzer Advia 2120 (Siemens Healthcare, Erlangen, Germany).

Recorded data were analyzed using Microsoft Office Excel 2007 and IBM SPSS 23 statistics programs. Data were described using frequencies, percentages, means and standard deviations where appropriate. Between-group differences were analyzed using the independent-samples T-test and Whitney U test, considering the normality of distribution (estimated by using the Shapiro–Wilk test). Univariate logistic regression analysis was performed to determine the parameters that can predict complicated and uncomplicated appendicitis occurrence, while multivariate analysis was applied to determine the overall impact. Discrimination between groups was estimated by performing the Receiver operating characteristics (ROC) analysis calculating the AUC (area under the curve), sensitivity and specificity for significant predictive parameters. A chi-square test was performed to determine differences in gender distribution between different groups. Calculated differences lower than the significance level of 0.05 were considered relevant.

## 3. Results 

During the two year period (Table 1), 223 appendectomies were performed at our Institute. There were 135 boys and 88 girls, with a median age of 11.02 (9.00–15.02) years. In 107 (47.98%) cases appendicitis was uncomplicated and in 116 (52.02%) it was complicated. We also analyzed blood tests of 239 patients treated at our Institute with nonspecific abdominal pain, as a control group (112 boys and 127 girls, with the median age of 12 (9.00–15.00) years). Significant differences between the operated patients and the control group were in sex, WBC and RDW/MPV ratio. In all other variables, the difference was not significant (*p* > 0.05). The only significant difference between the subgroups of operated patients was in WBC (*p* < 0.001) and the RDW/RBC ratio (*p* < 0.05).

Performing logistic regression analysis, laboratory parameter WBC and PDW/MPV ratio were found to be significant independent predictors for discriminating between patients in the control group and those with acute appendicitis. In patients with complicated and uncomplicated appendicitis, significant independent predictors are WBC and RDW/RBC. In multivariate logistic regression analysis, a significant joint effect of WBC and RDW/RBC was observed in the group of children with complicated appendicitis (Table 2). Only statistically significant predictors are presented.

By using the Receiver operating characteristics (ROC) analysis we assessed the discrimination of significant factors presented in Table 3 and Figure 1. WBC and RDW/MPV ratio with the area under the curve of 0.749 and 0.642 and cut-off values of 12.86 and 1.64 respectively separate the patients with acute appendicitis from the control group well enough. A higher percentage of specificity was seen in both variables in relation to sensitivity. The WBC and RDW/RBC ratio with the area under the curve of 0.684 and 0.633 and cut-off values of 15.05 and 2.50, respectively, separates patients with complicated from uncomplicated appendicitis well. In the RDW/RBC ratio, sensitivity is greater than specificity. WBC better separates patients with acute appendicitis than patients with complicated appendicitis. 

## 4. Discussion

Acute appendicitis is the most common acute surgical disease and cause of acute abdomen in patients aged between 10 and 20 years. Rapid and accurate diagnosis of acute appendicitis is particularly important. Diagnosis of appendicitis in children is challenging because of difficult communication, atypical presentation and conditions that mimic appendicitis. Many diagnostic tests are being investigated in order to facilitate the diagnosis. Avoiding perforation and subsequent complications must be weighed against the removal of a normal appendix. Negative appendectomy still occurs in 1 to 40% of patients. [3,19]. The rate of complicated appendicitis has been quoted as 20–74%, being even higher in younger children. The perforation rate in children is not associated with a delay in presentation. Barriers to health care access, longer duration of symptoms and the presence of an appendicolith have been associated with higher perforated appendicitis rates [20,21,22]. 

To date, several studies have focused on the diagnostic value of laboratory biomarkers as laboratory parameter of sufficient predictive value for the diagnosis of appendicitis is lacking. These tests should be easily accessible, reproducible, minimally invasive and cheap. However, the most commonly used laboratory tests are the number of leukocytes, absolute neutrophil count, and C-reactive protein (CRP) [23,24].

Many studies in adults have shown that, if symptoms and clinical examination clearly indicate the existence of acute appendicitis, the value of laboratory findings does not preclude the need for surgical treatment. This is the same in children. Serum inflammatory markers are age-dependent, so the results of studies in adults could not be administered to children. Some studies have demonstrated that CRP is more sensitive in determining the existence of perforated appendicitis, while the elevated white blood cell count is more sensitive in deciding whether a patient has or does not have acute appendicitis [23,24,25,26].

Recently, newer laboratory markers have been evaluated to provide accurate and timely diagnosis [8,12,13,14,27]. These include phospholipase A2, serum amyloid A, leukocyte elastase, interleukins and cytokines and tumor markers (CA-125). 

The MPV, PDW and RDW have been among the markers studied as diagnostic tests for acute appendicitis. These tests are simple and automatically measured as a part of routine complete blood count. However, they are not specific in the distinctive diagnosis of appendicitis, and may also increase in other inflammatory conditions.

In our study, WBC and PDW/MPV levels were significantly higher in patients with appendicitis. RBC and RDW/RBC levels were lower in patients with acute appendicitis compared with the control group. Significantly higher mean WBC and RDW/RBC ratios were found in patients with complicated appendicitis, and mean PDW values were lower in patients with complicated appendicitis.

Some studies showed that elevated levels of RDW can be seen in cases of acute appendicitis. A study performed by Narci et al. found that in adult patients with acute appendicitis, the RDW level is lower than in healthy population [7]. Tanrikulu et al. reported that there was no difference between acute appendicitis and the control group in terms of RDW [28]. Bozlu et al. showed that the value of RDW in children with acute appendicitis was higher. Additionally, they proved that this value is of little diagnostic significance in relation to the number of leukocytes and CRP [8]. The results of this study showed no significant difference in RDW values either between patients with acute appendicitis and the control group, or between patients with complicated and uncomplicated appendicitis.

MPV, a marker of platelet activation, is being investigated for its correlation with inflammation [29]. A decrease in MPV usually occurs in the acute phase, whereas an increase occurs with chronic events. Dinc et al. showed the diagnostic value of MPV in determining the existence of acute appendicitis in adults [5]. Hagi et al. concluded that MPV and RDW indexes have the potential to be used by surgeons in the diagnosis of acute and perforated appendicitis, especially in adults. They also emphasized that MPV is more valid in screening acute appendicitis, compared to the RDW [6]. In our study, no significant association was found between MPV and pediatric appendicitis, which is in agreement with the results obtained by Benito et al. who also did not find a difference in MPV values in children with appendicitis and those without appendicitis [30].

Studies in children with acute appendicitis have shown different results. Uyanik et al. confirm that MPV in children with acute appendicitis has no diagnostic value [18]. However, Oktay et al. found that low MPV showed importance for the diagnosis of acute appendicitis [31]. Bilici et al. also reported a significant reduction in MPV in the group of children with acute appendicitis in comparison with the control group [32].

PDW is an indicator of variation in platelet size, which can be a sign of active platelet release. Studies have demonstrated that, in addition to MPV, PDW is also altered compared to healthy subjects in several conditions. The study of Dinc et al. showed high values of sensitivity, specificity, and diagnostic accuracy of these parameters in patients with acute appendicitis [5]. Results of Sucu et al. suggest that platelet distribution width and mean platelet volume may be used for the diagnosis of appendicitis in children with the sensitivity of at least 77.6% and 78.1%, respectively [33]. In this study, PDW has no discriminant power to separate patients with appendicitis and patients with complicated appendicitis.

Up to now, most studies have confirmed the low accuracy of these diagnostic tests. Some researchers have tried to increase their sensitivity and specificity and combined them simultaneously [6].

Daldal and Dagmura evaluated in their study neutrophil-to-lymphocyte ratio (NLR) and platelet to lymphocyte ratio and (PLT/L) to support the diagnosis of acute appendicitis. They found the NLR values to be significantly higher in patients with acute appendicitis [34]. 

This study found that a WBC greater than 12.86 could be a new index for diagnosing acute appendicitis with the sensitivity of 66.3% and specificity of 73.2%, and values greater than 15.05 could diagnose complicated appendicitis with the sensitivity of 60.5% and specificity of 70.7%.

Hence, we tried to combine some of these tests and analyze the values of the ratio RDW/MPV, MPV/PDW and RDW/RBC. Our results showed that the RDW/MPV ratios differed significantly between patients in different groups, while the RDW/RBC ratio differed significantly in patients with complicated and uncomplicated appendicitis. RDW/MPV significantly discriminates patients with acute appendicitis, and RDW/RBC discriminates patients with complicated appendicitis. RDW/MPV ratios greater than 1.64 could be an index for diagnosing acute appendicitis with the sensitivity of 59.8% and specificity of 71.9%, and MDW/PDW ratio values greater than 2.50 could diagnose complicated appendicitis with the sensitivity of 72% and specificity of 52.8%.

Limitations of the study lie the fact that it presents a retrospective review and is a single-center design. Furthermore, these results should be reproduced in larger, prospective, randomized, multi-center trial in order to determine its significance in terms of diagnostics of acute appendicitis.

## 5. Conclusions

Our study showed that only WBC and RDW/MPV have diagnostic value in pediatric acute appendicitis, and WBC and RDW/RBC in predicting the degree of appendicitis. 

Further randomized studies on a larger number of patients are needed in order to determine its significance in terms of diagnostics of acute appendicitis.

## Figures and Tables

**Figure 1 children-08-01041-f001:**
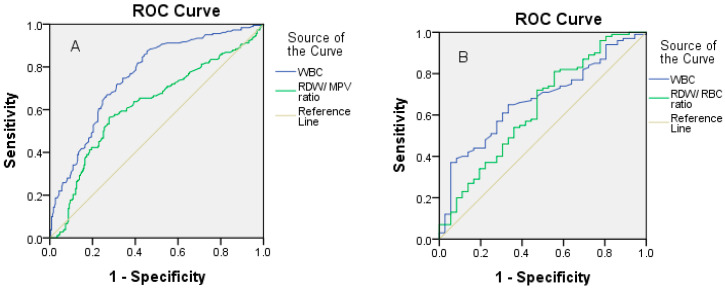
Receiver operating characteristic curves: (**A**)—for WBC and RDW/MPV ratio in predicting acute appendicitis; (**B**)—for WBC and RDW/RBC ratio in predicting complicated appendicitis.

**Table 1 children-08-01041-t001:** Demographic and laboratory data within groups and subgroups.

	Acute Appendicitis (*n* = 223)			Control Group (*n* = 239)	*p*-Value	
	Complicated (*n* = 116)	Uncomplicated (*n* = 107)	Overall		Groups	Subgroups
Male/female ^#^ (*n*)	74 /42	61/46	135/88	112/127	0.004	0.369
Age (y) ^†^	12.01 (9.00–14.04)	11.04 (8.03–15.02)	11.02 (9.00–15.02)	12 (9.00–15.00)	0.201	0.851
RBC (10^12^/L) ^†^	4.84 (4.54–5.06)	4.81 (4.53–5.12)	4.82 (4.53–5.11)	4.76 (4.48–5.01)	0.201	0.770
RDW (%) ^‡^	12.96 ± 1.31	12.95 ± 1.57	12.96 ± 1.43	12.95 ± 1.37	0.912	0.935
MPV (fl) ^‡^	7.49 ± 1.16	7.51 ± 0.56	7.49 ± 1.06	7.68 ± 0.87	0.053	0.942
PDW ^†^	17.15 (16.32–17.80)	17.20 (16.40–17.70)	17.19 (16.40–17.80)	17.30 (16.70–17.72)	0.336	0.719
WBC (10^9^/L) ^†^	16.10 (12.02–20.70)	11.40 (5.10–14.70)	14.20 (9.80–17.90)	9.50 (7.60–13.20)	<0.001	<0.001
RDW/MPV ratio ^†^	1.71 (1.50–2.06)	1.68 (1.55–1.90)	1.69 (1.53–1.98)	1.57 (1.46–1.68)	<0.001	0.924
RDW/RBC ratio ^†^	2.63 (2.46–2.89)	2.49 (2.33–2.76)	2.61 (2.43–2.86)	2.58 (2.44–2.80)	0.485	0.018
MPV/PDW ratio ^†^	0.46 (0.44–0.50)	0.46 (0.44–0.48)	0.46 (0.44–4.50)	0.49 (0.43–0.49)	0.144	0.694

^#^ Values are numbers; Chi-square test. ^†^ Values are median (interquartile range: Q1–Q3); Mann-Whitney U test. ^‡^ Values are means ± standard deviation; Independent-samples T-test. RBC—Red blood cell. RDW—Red blood cell distribution width. MPV—Mean platelet volume. PDW—Platelet distribution width. WBC—White blood cell.

**Table 2 children-08-01041-t002:** Logistic regression analysis between groups and subgroup.

		Univariate Analysis				Multivariate Analysis			
				95% CI for Exp (B)				95% CI for Exp (B)	
		*p*	Exp (B)	Lower	Upper	*p*	Exp (B)	Lower	Upper
Group	WBC (10^9^/L) ^†^	0.000	1.078	1.043	1.114				
	RDW/MPV ratio	0.023	1.098	1.002	1.103				
Subgroup	WBC (10^9^/L) ^†^	0.000	1.167	1.108	1.229	0.005	1.137	1.040	1.244
	RDW/RBC ratio ^†^	0.013	5.531	1.433	21.343	0.012	5.684	1.460	22.135

WBC ^†^—White blood cell. RDW/MPV ratio—Red Blood Cell Distribution Width/Mean platelet volume ratio. RDW/RBC ratio ^†^—Red Blood Cell Distribution Width/Red Blood Cell ratio.

**Table 3 children-08-01041-t003:** Proposed cut-off values for significant parameters in the diagnosis of acute appendicitis, and complicated appendicitis.

	AUC	Std. Error	95% CI	Cut-Off Value	Sensitivity %	Specificity %
Acute appendicitis						
WBC (10^9^/L) ^†^	0.749	0.023	0.704–0.795	12.86	66.3	73.2
RDW/MPV ratio ^†^	0.642	0.027	0.589–0.695	1.64	59.8	71.9
Complicated appendicitis						
WBC (10^9^/L) ^†^	0.684	0.038	0.609–0.758	15.05	60.5	70.7
RDW/RBC ratio ^†^	0.633	0.056	0.523–0.743	2.50	72.0	52.8

AUC—Area under the curve. WBC ^†^—White blood cell. RDW/MPV ratio ^†^—Red Blood Cell Distribution Width/Mean platelet volume ratio. RDW/RBC ratio—Red Blood Cell Distribution Width/Red Blood Cell ratio.

## Data Availability

The data presented in this study will be provided on request.

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
