# Peer review of "Predictive Value of Red Blood Cell Distribution Width, Mean Platelet Volume and Platelet Distribution Width in Children with Acute Appendicitis"

_children, 2021, doi:10.3390/children8111041_

Round 1
Reviewer 1 Report
Authors examined the predictive value of RDW, MPV and PDW in children
with acute appendicitis.
Although this manuscript is potentially interesting, several issues arise.
Almost physician believe that white blood cell (WBC) count and parameters of WBC are important to diagnose acute appendicitis. Authors should add WBC and parameters of WBC. Then, authors should compare WBC and its parameters with RDW, MPV, and PDW.
These data may not be normal distribution.
What is the definition of complicated or uncomplicated appendicitis?
Why did authors classify to complicated or uncomplicated appendicitis?
Why did authors perform the statistical analysis among 3 groups instead of 2 groups?
ROC analysis may be helpful.
The number of RDW, MPV and PDW are calculated and those ratio may increase statistical error.
Author Response
Point 1. Almost physician believe that white blood cell (WBC) count and parameters of WBC are important to diagnose acute appendicitis. Authors should add WBC and parameters of WBC. Then, authors should compare WBC and its parameters with RDW, MPV, and PDW.
Response 1: We did not include white blood cell count in our research because it has already been proven that the number of leukocytes is an important parameter in the algorithm for the diagnosis of acute appendicitis and already included in number of scoring systems.
Point 2. What is the definition of complicated or uncomplicated appendicitis?
Response: A clinical classification is used to stratify management based on simple, uncomplicated (non-perforated) and complex or complicated (gangrenous or perforated) inflammation. (Bhangu A, Soreide K, Di Saverio S, Assarsson JH, Drake FT. Acute appendicitis: modern understanding of pathogenesis, diagnosis, and management. Lancet. 2015;386(10000):1278-87.)
Point 3. Why did authors classify to complicated or uncomplicated appendicitis?
Why did authors perform the statistical analysis among 3 groups instead of 2 groups?
Response: In the study, we classified appendicitis into complicated and uncomplicated in order to try to determine whether any of these parameters can make a distinction between these two groups and thus determine the treatment strategy.
Therefore, we analyzed three instead of two groups.
Point 4. ROC analysis may be helpful.
Response: We will include ROC analysis.
Reviewer 2 Report
In their study Antic et al have evaluated the predictive value of red blood cell distribution width (RDW) and platelet morphology (volume MPV and variance PDW) as potential markers of acute appendicitis in pediatric patients. The authors mentioned correctly that all these parameters may change nonspecifically as a result of acute or chronic inflammatory process or tissue damage. The retrospective study included a cohort of patients with appendicitis (both complicated and uncomplicated) that was compared with a gender and age-matched cohort of pediatric patients with light head trauma. Choosing such a control cohort leaves the reader with the question, what we want to discriminate for wit the new potential predicting indicators. Is this a group with appendicitis against other inflammatory diseases or any kind of inflammatory disease with non-inflammatory RBC and platelet phenotypes? As rightfully stated, inflammation is a hallmark of appendicitis, but this is also true for any kind of inflammatory process (intestinal and non-intestinal), for which neutrophils and CRP would do the job. Why do we need RDW/PDW-based markers in addition to those known ones, what do they add? If you want to predict the severity of patients’ state with already clear diagnosis: appendicitis, it may be an additional parameter. Why are the other inflammatory markers not included into the PCA and correlative analysis, and not even shown in this paper? Is it not useful to choose a control group with patients diagnosed with e.g. allergy/asthma or other pathologies related to inflammation, and not the minor head trauma? Such comparison could show whether the changes in RDW or PDW or derivatives therefrom are specifically pointing to appendicitis and not at the inflammatory process as such. Please, clarify the reasoning behind such a study design.
Data presentation in Fig1 and 2 are not adequate. You cannot connect these “points”, they illustrate completely independent conditions and not a continuous process. Showing points per condition and not e.g. box and whiskers plots tells the reader nothing about the datasets you present here. What are the medians, what is a variance of the datasets, are they normally distributed, are there outliers?
Finally, the feasibility of derivatives (RDW/MPV and MPV/PDW) as potential markers was proposed as a conclusion for the obtained results. What do these ratios stand for physiologically? It is rather important to understand that to propose the limitations and correct use of these parameters in the future.
Author Response
Point 1. Choosing such a control cohort leaves the reader with the question, what we want to discriminate for wit the new potential predicting indicators. Is this a group with appendicitis against other inflammatory diseases or any kind of inflammatory disease with non-inflammatory RBC and platelet phenotypes? As rightfully stated, inflammation is a hallmark of appendicitis, but this is also true for any kind of inflammatory process (intestinal and non-intestinal), for which neutrophils and CRP would do the job. Why do we need RDW/PDW-based markers in addition to those known ones, what do they add?
Response: There are studies that suggest the importance of the parameters examined in this study, but often contradictory. Our idea was to analyze these simple tests that are used every day and are very accessible. These tests might provide the possibility of differentiation even acute appendicitis.
Point 2. If you want to predict the severity of patients’ state with already clear diagnosis: appendicitis, it may be an additional parameter.
Response: Certainly, these are just additional parameters. Our analysis does not exclude the use of different, already known parameters of inflammation. Our analysis does not include the use of some new laboratory parameters.
Point 3. Why are the other inflammatory markers not included into the PCA and correlative analysis, and not even shown in this paper?
Response: We did not include other inflammatory markers in our research because it has already been proven that these inflammatory markers are used in the algorithm for the diagnosis of acute appendicitis and are already included in number of scoring systems.
Point 4. Is it not useful to choose a control group with patients diagnosed with e.g. allergy/asthma or other pathologies related to inflammation, and not the minor head trauma? Such comparison could show whether the changes in RDW or PDW or derivatives therefrom are specifically pointing to appendicitis and not at the inflammatory process as such. Please, clarify the reasoning behind such a study design.
Response: We chose this control group because we routinely use a complete blood count in these patients, and they do not have inflammation. In the next study, we would choose the proposed criterion for the control group.
Reviewer 3 Report
The authors evaluated the predictive values of Red blood cell distribution width (RDW), Mean platelet volume (MPV) and Platelet distribution width (PDW) in children with acute appendicitis. They concluded that RDW, MPV and PDW alone have no diagnostic value in pediatric acute appendicitis or predicting the degree of appendix inflammation, but they pointed that RDW/MPV ratio can be useful predictor of appendix inflammation, with higher values in patients with more severe stage of disease.
Study is interesting, but a significant amount of changes are required before possible acceptance. I have several suggestions for improving the study:
- Introduction is poor and should be improved. The authors stated that a number of studies on adult patients which introduce new markers in diagnosis of acute appendicitis have been performed lately. This is not correct. Several studies have been performed in the pediatric population. Please revise!
- Next, the authors should present inflammatory and non-inflammatory markers in more detail in the introduction. Please add several sentences and references describing listed markers that should be used for acute appendicitis:
- CRP and immature granulocytes: Utility of biomarkers in predicting complicated appendicitis: can immature granulocyte percentage and C-reactive protein be used? Postgrad Med. 2021;133(7):817-821. doi: 10.1080/00325481.2021.1948306.
- Platelets: Role of platelet indices as a biomarker for the diagnosis of acute appendicitis and as a predictor of complicated appendicitis: A meta-analysis. Ann Med Surg (Lond). 2021;66:102448. doi: 10.1016/j.amsu.2021.102448.
- Procalcitonin, D-dimer, interleukin: Diagnostic efficacy of serum procalcitonin, IL-6, IL-2, and D-dimer levels in an experimental acute appendicitis model. Turk J Gastroenterol. 2019 Jul;30(7):641-647. doi: 10.5152/tjg.2019.18534.
- Bilirubin: Hyperbilirubinemia as an Indicator of Perforated Acute Appendicitis in Pediatric Population: A Prospective Study. Surg Infect (Larchmt). 2021 doi: 10.1089/sur.2021.107.
- 5-hydroxyindole acetic acid: Does elevated urinary 5-hydroxyindole acetic acid level predict acute appendicitis in children? Emerg Med J. 2016;33(12):848-852. doi: 10.1136/emermed-2015-205559.
- WBC, PDW, RDW, MPV: Are average platelet volume, red cell distribution width and platelet distribution width guiding markers for acute appendicitis treatment options? Int J Clin Pract. 2021;75(7):e14232. doi: 10.1111/ijcp.14232. Epub 2021 Apr 26.
- Fibrinogen: Plasma Fibrinogen: An Independent Predictor of Pediatric Appendicitis. J Indian Assoc Pediatr Surg. 2021;26(4):240-245. doi: 10.4103/jiaps.JIAPS_123_20.
- Sodium: Hyponatremia as a predictor of perforated acute appendicitis in pediatric population: A prospective study. J Pediatr Surg. 2021;56(10):1816-1821. doi: 10.1016/j.jpedsurg.2020.09.066.
- Please include information on the exact study period. The authors stated ‘’from 2019 to 2020’’. Please include months.
- Why did the authors include the patients with minor head trauma as a control group? It should be more appropriate to include the patients with non-specific abdominal pain because in everyday practice we are distinguishing between surgical and non-surgical abdominal conditions (not minor head trauma)?
- Exclusion criteria were not even mentioned. Please describe clear inclusion / exclusion criteria of the study.
- There are no clear primary / secondary outcomes of the study. Please provide a separate paragraph in methodology with hypothesis, primary and secondary outcome measurements of this study!
- The authors stated that all patients were operated by open or laparoscopic technique. Please provide a description of the technique or include adequate reference.
- Which statistical test was used to evaluate normality of distribution?
- Results: In my opinion the following text should be deleted. ‘’ This section may be divided by subheadings. It should provide a concise and precise description of the experimental results, their interpretation, as well as the experimental conclusions that can be drawn.’’ What did the authors want to point out in this description? It is not appropriate.
- The authors should avoid statements such as ‘’ We have also analyzed blood tests…’’ Better construction would be: ‘’The blood tests were analyzed…’’. Please revise through the text.
- Resolution of Figures 1 and 2 is very poor. Please provide Figures with better resolution. This is very hard to follow.
- Discussion – The authors stated ‘’Recently, newer laboratory markers have been evaluated such as phospholipase A2, serum amyloid A, leukocyte elastase, interleukins and cytokines and tumor markers (CA-125) to provide accurate and timely diagnosis [16].’’ They provided only reference for CA-125. Adequate references for other mentioned markers should be cited. These markers should be mentioned in the introduction as well.
- Discussion should be improved. In this section the authors should provide interpretation and explanation of their results compared to the literature. The authors should focus on results from the main objectives of the study. Please avoid repetition of well-known facts from the literature. Discussion should be rearranged as follows: (a) summary (not data) of findings from present study; (b) logical and coherent comparison with existing literature with focus of comparison on main objective(s); (c) limitations of the study and implications for practice/policy/research with a concluding statement.
- The authors did not even mention limitations of their study. This should be included at the end of discussion
- Quality of English should be improved. Manuscript should be edited for the language from a native English speaker or professional language editing service to improve the grammar and readability.
If the authors are willing to put in the effort and rewrite the work according to the instructions I will be happy to review it again
Author Response
Point 1. Introduction is poor and should be improved. The authors stated that a number of studies on adult patients which introduce new markers in diagnosis of acute appendicitis have been performed lately. This is not correct. Several studies have been performed in the pediatric population. Please revise!
Response: Thanks for the suggestion. We will accept it and include studies on pediatric patients in the introduction.
Point 2. Next, the authors should present inflammatory and non-inflammatory markers in more detail in the introduction. Please add several sentences and references describing listed markers that should be used for acute appendicitis.
Response: Thanks for the suggestion. We will add several sentences and references describing listed markers that should be used for acute appendicitis.
Point 3. Please include information on the exact study period. The authors stated ‘’from 2019 to 2020’’. Please include months.
Response: Thanks for the suggestion. We will include exact study period.
Point 4. Why did the authors include the patients with minor head trauma as a control group? It should be more appropriate to include the patients with non-specific abdominal pain because in everyday practice we are distinguishing between surgical and non-surgical abdominal conditions (not minor head trauma)?
Response: We included patients with minor head trauma as a control group because the reason for non-specific abdominal pain is often some kind of inflammation. It is known that there is a change in the examined parameters in various inflammatory diseases. There is no inflammation in patients with minor head trauma.
Point 5. Exclusion criteria were not even mentioned. Please describe clear inclusion / exclusion criteria of the study.
Response: Thanks for the suggestion. We will include inclusion / exclusion criteria of the study.
Point 6. There are no clear primary / secondary outcomes of the study. Please provide a separate paragraph in methodology with hypothesis, primary and secondary outcome measurements of this study!
Response: Thanks for the suggestion. We will provide a separate paragraph in methodology with hypothesis, primary and secondary outcome.
Point 7. The authors stated that all patients were operated by open or laparoscopic technique. Please provide a description of the technique or include adequate reference.
Response: We did not describe the technique of the operation because it is not the subject of research, but we will describe both techniques.
Point 8. Which statistical test was used to evaluate normality of distribution?
Response: To evaluate normality of distribution, it was used Kolmogorov- Smirnov test.
Point 9. Results: In my opinion the following text should be deleted. ‘’ This section may be divided by subheadings. It should provide a concise and precise description of the experimental results, their interpretation, as well as the experimental conclusions that can be drawn.’’ What did the authors want to point out in this description? It is not appropriate.
Response: Thanks for the suggestion. We will delete this part of text.
Point 10. The authors should avoid statements such as ‘’ We have also analyzed blood tests…’’ Better construction would be: ‘’The blood tests were analyzed…’’. Please revise through the text.
Response: Thanks for the suggestion. We will change these constructions.
Point 11. Resolution of Figures 1 and 2 is very poor. Please provide Figures with better resolution. This is very hard to follow.
Response: Thanks for the suggestion. We will change resolution of these Figures.
Point 12. Discussion – The authors stated ‘’Recently, newer laboratory markers have been evaluated such as phospholipase A2, serum amyloid A, leukocyte elastase, interleukins and cytokines and tumor markers (CA-125) to provide accurate and timely diagnosis [16].’’ They provided only reference for CA-125. Adequate references for other mentioned markers should be cited. These markers should be mentioned in the introduction as well.
Response: Thanks for the suggestion. We will cite adequate references.
Point 13. Discussion should be improved. In this section the authors should provide interpretation and explanation of their results compared to the literature. The authors should focus on results from the main objectives of the study. Please avoid repetition of well-known facts from the literature. Discussion should be rearranged as follows: (a) summary (not data) of findings from present study; (b) logical and coherent comparison with existing literature with focus of comparison on main objective(s); (c) limitations of the study and implications for practice/policy/research with a concluding statement.
Response: Thanks for the suggestion. We will try to improve it.
Point 14. The authors did not even mention limitations of their study. This should be included at the end of discussion
Response: Thanks for the suggestion. We will include limitations of our study at the end of discussion.
Round 2
Reviewer 1 Report
Revised manuscript has been partially improved. However, several issues remain.
If the usefulness of WBC has been established in appendicitis, authors should compare the usefulness of new data with that of WBC.
Physician usually see WBC, RDW, MPV, and PDW in laboratory data at same time.
Authors should examine the usefulness of new data with WBC.
The deleted data and letter should be disappeared and only additional letters should be highlighted, because revised manuscript will be complicated.
Author Response
Response to Reviewer 1 Comments
Point 1
If the usefulness of WBC has been established in appendicitis, authors should compare the usefulness of new data with that of WBC.
Physician usually see WBC, RDW, MPV, and PDW in laboratory data at same time.
Authors should examine the usefulness of new data with WBC.
Response 1: We followed the suggestion and compared the usefulness of new data with that of WBC.
Point 2
The deleted data and letter should be disappeared and only additional letters should be highlighted, because revised manuscript will be complicated.
Response 2: The changes in the work are significant and we hope that they will be clearly seen
Reviewer 2 Report
Some of the aspects of the paper improved a lot, such as the quality of figures and the table with the overview of the findings in the area are very good. Discussion is still a bit of a mess. Some comments are in the attached text

Author Response
Point 1. Some of the aspects of the paper improved a lot, such as the quality of figures and the table with the overview of the findings in the area are very good. Discussion is still a bit of a mess. Some comments are in the attached text
Response: The discussion was corrected. But,I'm sorry, but the study in the attachment doesn't look like our study.
Reviewer 3 Report
The authors improved the manuscript significantly. However, some objections still remains open:
- First and the most important objection which has not been resolved is study design. In my opinion control group is not adequate. In everyday practice we do not distinguish between the patients with suspicion of acute appendicitis and minor head trauma. We have to distinguish between the patients with non-specific abdominal pain (which does not require surgery) and acute appendicitis. So, what is clinical value of this paper if we cannot implement it in everyday practice? The control group should be a patients with non-specific abdominal pain and if the patients with acute appendicitis really have increased values compared to non-specific abdominal pain group then this makes sense. This is one of the biggest limitations of the study and should be a significant source of bias.
- Introduction – ‘’Several studies have been performed in the pediatric population as well’’. Please provide adequate reference(s). The same goes for a sentence: ‘’However, in some cases, none of the new inflammatory markers, including leukocyte, C-reactive protein (CRP), or PCT, can identify acute appendicitis with high specificity and sensitivity’’
- In the revised manuscript the authors stated that Shapiro–Wilk test was used to test normality of distribution wile in a response to my objections they stated that Kolmogorov- Smirnov test was used. Please revise and state in manuscript correct information.
- Statistical analysis – For each abbreviation full title should be provided when the abbreviation is first time mentioned. Please provide full title for ‘’ROC’’ - Receiver operating characteristic curve.
- Tables 1-3 – Presentation of variables is incomplete: Near each variable it should be clearly stated whether it is n (%), median (IQR) or mean±SD
- In each table all of the abbreviations should be mentioned in legend whether or not already mentioned in the previous tables or text.
- I am happy that the authors performed ROC analysis and significantly improved the quality of manuscript. However, ROC curve (graph) should be included in this manuscript.
- I am still not happy with the organization of the discussion. As I pointed in previous review discussion section needs to be re-arranged as follows: (i) summary (not data) of findings from present study; (ii) logical and coherent comparison with existing literature with focus of comparison on main objective(s); (iii) limitations of the study; and (iv) Implications for practice/policy/research with a concluding statement.
- The authors ignored my comment that the quality of English should be improved.
I think the manuscript is valuable, but in this form it is still not acceptable. I hope that the authors will revise the manuscript according to the above remarks. I will be happy to review the revised version again
Best regards
John
Author Response
Response to Reviewer 2 Comments
Point 1. First and the most important objection which has not been resolved is study design. In my opinion control group is not adequate. In everyday practice we do not distinguish between the patients with suspicion of acute appendicitis and minor head trauma. We have to distinguish between the patients with non-specific abdominal pain (which does not require surgery) and acute appendicitis. So, what is clinical value of this paper if we cannot implement it in everyday practice? The control group should be a patients with non-specific abdominal pain and if the patients with acute appendicitis really have increased values compared to non-specific abdominal pain group then this makes sense. This is one of the biggest limitations of the study and should be a significant source of bias.
Response 1: We changed the control group and analyzed patients with nonspecific abdominal pain.
Point 2. Introduction – ‘’Several studies have been performed in the pediatric population as well’’. Please provide adequate reference(s). The same goes for a sentence: ‘’However, in some cases, none of the new inflammatory markers, including leukocyte, C-reactive protein (CRP), or PCT, can identify acute appendicitis with high specificity and sensitivity’’
Response 2: We provided adequate references.
Point 3. In the revised manuscript the authors stated that Shapiro–Wilk test was used to test normality of distribution wile in a response to my objections they stated that Kolmogorov- Smirnov test was used. Please revise and state in manuscript correct information.
Response 3: Sorry, it was a mistake. Shapiro–Wilk test was used to test normality of distribution. We stated correct information.
Point 4. Statistical analysis – For each abbreviation full title should be provided when the abbreviation is first time mentioned. Please provide full title for ‘’ROC’’ - Receiver operating characteristic curve.
Response 4: - We changed it. For each abbreviation full title is provided.
Point 5. Tables 1-3 – Presentation of variables is incomplete: Near each variable it should be clearly stated whether it is n (%), median (IQR) or mean±SD
Response 5: - We made correction.
Point 6. In each table all of the abbreviations should be mentioned in legend whether or not already mentioned in the previous tables or text.
Response 6: We made correction.
Point 7. I am happy that the authors performed ROC analysis and significantly improved the quality of manuscript. However, ROC curve (graph) should be included in this manuscript.
Response 7: We put ROC curve in this manuscript.
Point 8. I am still not happy with the organization of the discussion. As I pointed in previous review discussion section needs to be re-arranged as follows: (i) summary (not data) of findings from present study; (ii) logical and coherent comparison with existing literature with focus of comparison on main objective(s); (iii) limitations of the study; and (iv) Implications for practice/policy/research with a concluding statement.
Response 8: We made correction with organization of the discussion.
Point 9. The authors ignored my comment that the quality of English should be improved.
Response 9: We hired professional translators to correct the English and they made correction.